# Query-Efficient Locally Private Hypothesis Selection via the Scheffe Graph*

**Gautam Kamath**
University of Waterloo and Vector Institute
g@csail.mit.edu

**Alireza F. Pour**
University of Waterloo
alireza.fathollahpour@uwaterloo.ca

**Matthew Regehr**
University of Waterloo
matt.regehr@uwaterloo.ca

**David P. Woodruff**
Carnegie Mellon University
dwoodruf@cs.cmu.edu

## Abstract

We propose an algorithm with improved query-complexity for the problem of hypothesis selection under local differential privacy constraints. Given a set of $k$ probability distributions $Q$, we describe an algorithm that satisfies local differential privacy, performs $\tilde{O}(k^{3/2})$ non-adaptive queries to individuals who each have samples from a probability distribution $p$, and outputs a probability distribution from the set $Q$ which is nearly the closest to $p$. Previous algorithms required either $\Omega(k^2)$ queries or many rounds of interactive queries. Technically, we introduce a new object we dub the Scheffé graph, which captures structure of the differences between distributions in $Q$, and may be of more broad interest for hypothesis selection tasks.

## 1 Introduction

Hypothesis selection refers to the following statistical question: given $n$ samples from a distribution $p$, and descriptions of $k$ distributions $Q$, output a distribution $\hat{q}$ which is as close to $p$ as possible. More precisely, for an $\alpha > 0$, the goal is to output a distribution $\hat{q}$ such that

$$\|\hat{q} - p\|_1 \le O(1) \cdot \min_{q \in Q} \|q - p\|_1 + \alpha.$$

In other words, the $\ell_1$-distance between $p$ and the output distribution $\hat{q}$ is at most a constant factor larger than that of the closest distribution $q^* \in Q$, up to some additional additive error $\alpha$. How many samples are needed for this task, and what algorithms do we use to do it? This fundamental primitive serves as an important building block for many other statistical estimation tasks. Furthermore, it generalizes one of the most classic problems in statistics, simple hypothesis testing, wherein the distribution $p$ is promised to be exactly equal to one of the distributions $q \in Q$.

Many classical works (e.g., [Yat85, DL96, DL97, DL01]) address and resolve these questions, showing that $n = O(\log k)$ samples suffice. That is, we require only *logarithmically*-many samples in order to identify the (near-)best distribution. Subsequently, many other works have studied hypothesis selection subject to other constraints and desiderata, including computational efficiency, robustness, weaker access to hypotheses, and more (see, e.g., [MS08, DDS12, DK14, SOAJ14, AJOS14, DKK+16, AFJ+18, BKM19, BKSW19, GKK+20]).

We focus on the constraint of differential privacy (DP) [DMNS06], a rigorous notion of data privacy that guarantees that a procedure will not leak too much information about individual points in the

---

*Authors are listed in alphabetical order.

39th Conference on Neural Information Processing Systems (NeurIPS 2025).

dataset. DP has been adopted in practice by numerous organizations, including Google [XZA+23], Apple [Dif17], and the US Census Bureau [AACM+22]. Under the *central* notion of DP, wherein there exists a trusted curator who may observe the sensitive dataset directly, Bun, Kamath, Steinke, and Wu showed that $O(\log k)$ samples still suffice to perform hypothesis selection [BKSW19, BKSW21].

However, central DP requires a trusted curator, a strong assumption when operating on sensitive data. Instead, one can consider *local* DP (LDP) [War65, EGS03, KLN+11]: in this setting, every dataholder makes their own outputs DP before sharing them with anyone else. This offers much stronger privacy semantics than central DP, but also requires more data for most tasks.

Work by Gopi, Kamath, Kulkarni, Nikolov, Wu, and Zhang [GKK+20] initiated the study of hypothesis section under local DP [GKK+20]. In this case, each user holds an independent sample from the unknown distribution $p$. Unfortunately, lower bounds of Duchi and Rogers for sparse mean estimation imply that $\Omega(k)$ samples are necessary for this problem [DR19], exponentially more than the $O(\log k)$ samples which suffice for the central DP setting.

With this barrier in mind, [GKK+20] proved two main results. First, they provided an $\tilde{O}(k)$-sample algorithm for locally private hypothesis selection. This matches the lower bound of Duchi and Rogers up to logarithmic factors, and subsequent work by Pour, Ashtiani, and Asoodeh improves the upper bound to $O(k)$ [PAA24], matching the lower bound up to constant factors. The major caveat of both these algorithms is that they require *interactivity*. This is because the specific queries asked to each dataholder depend on the outputs provided by earlier dataholders (i.e., the queries are selected adaptively). This style of interactivity can be a non-starter for real-world deployments of local DP. If one desires a *non-interactive* algorithm, a straightforward privatization of the celebrated Scheffé tournament requires $O(k^2)$ samples. [GKK+20] improve upon this with their second main result, a non-interactive $\tilde{O}(k)$-sample algorithm, but for the simpler problem of $k$-wise simple hypothesis testing, where the distribution $p$ is promised to be *equal* to one of the distributions $q \in Q$.[2]

To summarize, we highlight three existing results under LDP:

- An interactive $O(k)$-sample algorithm for hypothesis selection;
- A non-interactive $O(k^2)$-sample algorithm for hypothesis selection; and
- A non-interactive $\tilde{O}(k)$-sample algorithm for simple hypothesis testing.

## 1.1 Results and Techniques

Our main result improves upon all of these, providing a non-interactive $\tilde{O}(k^{3/2})$-sample algorithm for LDP hypothesis selection, where $\tilde{O}(f) = f \cdot \text{polylog}(f)$. Definitions are given in Section 2.

**Theorem 1.** *Given a set of $k$ distributions $Q$ and $\tilde{O}(k^{5/2})$ expected preprocessing time[3], there exists a non-interactive $\varepsilon$-locally differentially private algorithm with the following guarantees. For any $\alpha, \beta > 0$, there is*

$$n_0 \leq O\left(\frac{k^{3/2}\sqrt{\log k}\log(k/\beta)}{\alpha^2\varepsilon^2}\right)$$

*such that given $n \geq n_0$ samples from a distribution $p$, then with probability at least $1 - \beta$ the algorithm outputs a distribution $\hat{q} \in Q$ satisfying*

$$\|\hat{q} - p\|_1 \leq 13 \cdot \min_{q \in Q}\|q - p\|_1 + \alpha.$$

In fact, by applying our algorithm recursively[4] as in [GKK+20], we also get a multi-round procedure with improved sample complexity.

---

[2]We simplify for the sake of presentation: they actually show a slightly stronger result. Let $OPT = \min_{q \in Q}\|q - p\|$. Roughly speaking, their Lemma 4.1 provides a non-interactive $\varepsilon$-LDP algorithm such that, given $n = \tilde{O}(k/\alpha^4\varepsilon^2)$ samples, it outputs a distribution $\hat{q}$ such that $\|\hat{q} - p\| \leq O(\sqrt{\log k}) \cdot \sqrt{OPT} + O(\alpha)$. Note that, compared to our desired $O(1) \cdot OPT$ guarantees, their result degrades quadratically in the value of OPT, and weakens as the number of hypotheses $k$ becomes large.

[3]Preprocessing involves computing many probabilities $q(E)$ for $q \in Q$, which we treat as constant-time.

[4]See Algorithm 6 in [GKK+20] Subsection 5.4. Corollary 2 follows immediately by applying their technique with their 1-round subroutine replaced by ours and using a slightly modified subdivision rate $\eta_t := 2/(3^t - 1)$ as well as a larger random set $|H| = O(k^{2 \cdot 3^{t-1}/(3^t-1)})$ in the final round to account for our improved base rate.

**Corollary 2.** *Given a set of $k$ distributions $Q$ and $\tilde{O}(k^{5/2})$ expected preprocessing time, there exists a $t$-round $\varepsilon$-locally differentially private protocol with the following guarantees. For any $\alpha, \beta > 0$, there is*

$$n_0 \leq O\left(\frac{k^{1+\frac{1}{3^t-1}} t \sqrt{\log k} \log(k/\beta)}{\alpha^2 \varepsilon^2}\right)$$

*such that given $n \geq n_0$ samples from a distribution $p$, then with probability at least $1 - \beta$ the algorithm outputs a distribution $\hat{q} \in Q$ satisfying*

$$\|\hat{q} - p\|_1 \leq 39 \cdot \min_{q \in Q}\|q - p\|_1 + \alpha.$$

To prove Theorem 1, we first introduce in Section 3 a generalization of the classical minimum distance estimator that accepts any collection of queries that is rich enough to facilitate comparisons between any pair of distributions. Next, we show in Section 4 how standard tools from differential privacy can be used to non-interactively estimate the queries under local privacy. Our final and most crucial step is to introduce in Section 5 a new combinatorial object, the Scheffé graph. This is a directed graph whose vertices correspond to possible queries that a locally private hypothesis selection algorithm may ask and whose directed edges indicate when one query gives sufficient information to answer another query. It is natural to ask for a minimal set of queries that yield sufficient information to answer all queries and we show that there indeed exists such a small such set of queries. Theorem 1 then follows immediately by combining Theorem 12 in Section 4 with Theorem 14 in Section 5 below.

The natural question is whether our bound can be strengthened to achieve an $\tilde{O}(k)$ sample complexity for non-interactive locally private hypothesis selection. A core part of our analysis involves showing an $\tilde{O}(k^{3/2})$ bound on the domination number of any Scheffé graph – if this could be improved to $\tilde{O}(k)$, then it would produce the desired result. However, in Section 6.1, we provide a nearly-matching lower bound on the domination number, showing that additional structure must be employed to go beyond this barrier.

Another approach to designing an $\tilde{O}(k)$ sample algorithm is based on a suggestion of [GKK+20]. A key technical component of their work is a so-called *flattening lemma* – they point out that a specific strengthening would lead to an $\tilde{O}(k)$ sample algorithm. In Section 6.2, we provide a concrete counterexample to such a strengthening, showing that it is not achievable.

## 1.2  Related Work

Hypothesis selection is a classical statistical task. This style of approach was introduced by Yatracos [Yat85], and further developed in subsequent work by Devroye and Lugosi [DL96, DL97, DL01]. The most relevant line of work to ours studies hypothesis selection under differential privacy constraints [BKSW19, AAK21, GLW21, GKK+23, PAA24]. However, all of these works either study a weaker notion of privacy, require interactivity, require more data, or apply to a weaker problem than our work. Another line of work focuses on algorithms for (non-private) hypothesis selection that minimize the number of comparisons or the amount of computation [MS08, DDS12, DK14, SOAJ14, AJOS14, AFJ+18, ABS24]. While many of these algorithm require only a near-linear number of comparisons between hypotheses, they are unsuitable for our purposes as they perform adaptive queries, which would result in an interactive protocol in our setting. There are a number of other works on hypothesis selection, focusing on desiderata including robustness [DKK+16, BBKL23], approximation factor [BKM19, BBK+22], memory constraints [ABS23], and more [QCR20, AAC+23, AAC+24]. There has also been significant work into hypothesis testing under local DP [DJW13, DJW17, GR18, She18, ACFT19, ACT19, JMNR19, AZ24, PAJL24, PJL24], though this often focuses on the non-agnostic case (i.e., when the distribution is exactly equal to one of the given distributions) and $k = 2$. For more coverage of private statistics, see [KU20].

## 2  Preliminaries

We recall the classic definitions of differential privacy (DP) and sequentially interactive local differential privacy (LDP):

**Definition 3** ([DMNS06])**.** *An algorithm $M : \mathcal{X}^n \to \mathcal{Y}$ is $\varepsilon$-differentially private if, for all $X, X' \in \mathcal{X}^n$ that differ in exactly one entry and $S \subseteq \mathcal{Y}$, we have that*

$$\Pr[M(X) \in S] \le e^\varepsilon \Pr[M(X') \in S].$$

**Definition 4** ([War65, EGS03, KLN$^+$11])**.** *A $t$-round sequentially interactive protocol is defined by the following interaction between users and a central server. Suppose there are $n$ users, where user $i \in \{1, \ldots, n\}$ has datapoint $X_i \in \mathcal{X}$. During round $j \in \{1, \ldots, t\}$ of communication, the central server transmits arbitrary messages to a new subset $U_j \subseteq \{1, \ldots, n\} \setminus \bigcup_{j'=1}^{j-1} U_{j'}$ of users in parallel. These users transmit randomized messages back to the server in parallel. The messages sent by the server may depend on messages it received during any of the previous rounds of communication and the messages sent by a user $i$ may depend on her datapoint $X_i$ as well as the message she received from the server. We say that the protocol is $\varepsilon$-locally differentially private (LDP) if the record of all messages in all rounds of communication transmitted from the users to the server is $\varepsilon$-DP with respect to the datapoints $(X_1, \ldots, X_n)$. If $t = 1$, we say that the protocol is non-interactive.*

We also recall the notion of a dominating set in a directed graph.

**Definition 5.** *Let $G = (V, E)$ be a digraph. A dominating set for $G$ is a subset $D \subseteq V$ of vertices such that that, for every vertex $w \in V$, either $w \in D$, or there is $v \in D$ such that $(v, w) \in E$, i.e. there is an edge $v \to w$. We call the size of a minimal dominating set the domination number of $G$, which we write as $dom(G)$.*

We will sometimes say a vertex $v$ dominates a set of vertices $W$, which means that for each $w \in W$, either $w = v$ or there is an edge $v \to w$. In the same vein, we say that a set of vertices $U$ dominates a set $W$ if each $w \in W$ is dominated by some $v \in U$.

Finally, we recall the classical Scheffé test. Note that we frequently conflate a distribution $q$ with its mass function (density in the continuous setting) to make expressions such as $q(x)$ and $\langle q, T \rangle$ legible.

**Definition 6.** *For a pair of distributions $q, q' \in \Delta(\mathcal{X})$ over a domain $\mathcal{X}$, we denote by $\delta(x) := q(x) - q'(x)$ the difference functional from $q$ to $q'$ and we denote by*

$$S(x) := sgn(\delta(x)) = \begin{cases} +1 & \text{if } q(x) \ge q'(x) \\ -1 & \text{if } q(x) < q'(x) \end{cases}$$

*the signed Scheffé set from $q$ to $q'$.*

In some sense, "querying" the signed Scheffé set $S$ is the best possible measure of the $\ell_1$ distance between $q$ and $q'$, formalized in the following lemma.

**Lemma 7.** *For any distributions $q$ and $q'$ with signed Scheffé set $S$,*

$$\|q - q'\|_1 = \langle \delta, S \rangle = \sup_{T \in \{-1, 1\}^{\mathcal{X}}} |\langle q - q', T \rangle|.$$

The classical Scheffé test between $q$ and $q'$ involves sampling data from some unknown distribution $p$, calculating an estimate $\hat{p}_S$ of $\langle p, S \rangle$, then returning $q$ if $\langle q, S \rangle$ is closer to $\hat{p}_S$ than $\langle q', S \rangle$ and returning $q'$ otherwise. This estimator can be shown [DL01] to obtain $\ell_1$-error at most

$$3 \min\{\|q - p\|_1, \|q' - p\|_1\} + 2|\langle p, S \rangle - \hat{p}_S|.$$

## 3 The Relaxed Minimum Distance Estimator

In this section, we develop an estimator for $k$ distributions with a similar guarantee to that of the classical Scheffé test. We assume that we are given access to some estimates $\hat{p}_T$ of $\langle p, T \rangle$ where $p$ is an unknown distribution and where $T$ belongs to a family of queries $\mathcal{T}$. Moreover, under the LDP constraints, each query $\hat{p}_T$ requires fresh data, so we would like some estimator that only makes a small number of distinct queries to $p$.

**Definition 8** (Relaxed Minimum Distance Estimator (RMDE))**.** *Let $Q \subseteq \Delta(\mathcal{X})$ be a finite set of distributions and suppose we have collection $\mathcal{T}$ of functionals $T \in \{-1, 1\}^{\mathcal{X}}$ as well as a sequence of query results $\hat{p}_{\mathcal{T}} = (\hat{p}_T)_{T \in \mathcal{T}}$. The relaxed minimum distance estimate given the query results is*

$$\hat{q}(\hat{p}_{\mathcal{T}}) := \underset{q \in Q}{\arg\min} \sup_{T \in \mathcal{T}} |\langle q, T \rangle - \hat{p}_T|.$$

The following theorem provides theoretical guarantees for the RMDE – similar to the Scheffé test, it can be decomposed into the error from the optimal hypothesis plus approximation error over the set of functionals $\mathcal{T}$.

**Theorem 9.** *Let $Q$ be a finite set of distributions over $\mathcal{X}$ and let $\mathcal{T} \subseteq \{-1, 1\}^{\mathcal{X}}$ be a set of functionals with the property that, for each $q, q' \in Q$, there is some $T \in \mathcal{T}$ satisfying*

$$|\langle q - q', T \rangle| \geq \phi \|q - q'\|_1. \tag{$\star$}$$

*Then, for any distribution $p$ and query results $\hat{p}_{\mathcal{T}} = (\hat{p}_T)_{T \in \mathcal{T}}$,*

$$\|\hat{q}(\hat{p}_{\mathcal{T}}) - p\|_1 \leq (1 + 2\phi^{-1})\|q^* - p\|_1 + 2\phi^{-1} \sup_{T \in \mathcal{T}} |\langle p, T \rangle - \hat{p}_T|$$

*where $q^* := \arg\min_{q \in Q} \|q - p\|_1$ denotes the closest distribution to $p$.*

One could take the query set $\mathcal{T}$ to be all $\binom{k}{2}$ signed Scheffé sets between pairs $q, q' \in Q$. This recovers the classical minimum distance estimator [DL01], and would satisfy the theorem condition with $\phi = 1$. Our goal will be to obtain a smaller query set $\mathcal{T}$ (which will translate into fewer queries and samples), at the cost of a smaller value of $\phi$.

*Proof.* Write $\hat{q} := \hat{q}(\hat{p}_{\mathcal{T}})$ for short. Clearly, $\|\hat{q} - p\|_1 \leq \|q^* - p\|_1 + \|\hat{q} - q^*\|_1$, so we will just focus on bounding $\|\hat{q} - q^*\|_1$.

Now, by $(\star)$, there must be some $\hat{T} \in \mathcal{T}$ for which

$$\begin{aligned}
\|\hat{q} - q^*\|_1 &\leq \phi^{-1} |\langle \hat{q} - q^*, \hat{T} \rangle| \\
&\leq \phi^{-1} \sup_{T \in \mathcal{T}} |\langle \hat{q} - q^*, T \rangle| \\
&\leq \phi^{-1} \left( \sup_{T \in \mathcal{T}} |\langle \hat{q}, T \rangle - \hat{p}_T| + \sup_{T \in \mathcal{T}} |\langle q^*, T \rangle - \hat{p}_T| \right) \\
&\leq 2\phi^{-1} \sup_{T \in \mathcal{T}} |\langle q^*, T \rangle - \hat{p}_T| \\
&\leq 2\phi^{-1} \left( \sup_{T \in \mathcal{T}} |\langle q^* - p, T \rangle| + \sup_{T \in \mathcal{T}} |\langle p, T \rangle - \hat{p}_T| \right) \\
&\leq 2\phi^{-1} \|q^* - p\|_1 + 2\phi^{-1} \sup_{T \in \mathcal{T}} |\langle p, T \rangle - \hat{p}_T|
\end{aligned}$$

where the last inequality follows from Lemma 7. $\square$

## 4 Non-Interactive Locally Differentially Private RMDE

In this section, we explain how to get accurate estimates of $\hat{p}_T$ of $\langle p, T \rangle$ under $\varepsilon$-LDP. Consequently, we will achieve non-interactive LDP hypothesis selection by calculating all of these estimates in parallel and then supplying them to the relaxed minimum distance estimator.

We first recall randomized response, which is a classical mechanism that ensures local privacy by flipping the response bit with small probability. This introduces a (correctable) bias.

**Lemma 10** ([War65, EGS03, KLN$^+$11]). *Randomized response is the randomized function $\mathsf{RR}_\varepsilon$ that receives $x \in \{-1, 1\}$ and outputs $x$ with probability $\frac{e^\varepsilon}{e^\varepsilon + 1}$ and $-x$ with probability $\frac{1}{e^\varepsilon + 1}$. Randomized response satisfies $\varepsilon$-LDP.*

Assuming each user holds an independent datapoint $x \sim p$, we can estimate our workload of queries $\langle p, T \rangle$ under LDP by applying randomized response to $T(x)$ for each user, averaging, and correcting the bias introduced by $\mathsf{RR}_\varepsilon$.

**Proposition 11.** *Let $\mathcal{T}$ be a collection of functionals $T \in \{-1, 1\}^{\mathcal{X}}$. Then there is an $\varepsilon$-LDP mechanism which requires $m = O\left(\frac{|\mathcal{T}| \log(|\mathcal{T}|/\beta)}{\alpha^2 \varepsilon^2}\right)$ samples and computes estimates $\hat{p}_{\mathcal{T}} = (\hat{p}_T)_{T \in \mathcal{T}}$ such that with probability at least $1 - \beta$, we have $|\langle p, T \rangle - \hat{p}_T| \leq \alpha$ for all $T \in \mathcal{T}$.*

*Proof.* Assume we have a sample $S$ from $p$ distributed locally among users. The curator divides the sample into $|\mathcal{T}|$ disjoint subsets $S_1, \ldots, S_{|\mathcal{T}|}$ each of size $\ell = |S|/|\mathcal{T}| = O\left(\frac{\log(|\mathcal{T}|/\beta)}{\alpha^2\varepsilon^2}\right)$. Fix an enumeration $\pi$ on the functionals in $\mathcal{T}$. For each $T \in \mathcal{T}$, every user in $S_{\pi(T)}$ with sample $x$ sends $m(x) := \mathsf{RR}_\varepsilon(T(x))$ to the curator, who computes $\hat{p}_T = \frac{1}{\ell} \cdot \frac{e^\varepsilon+1}{e^\varepsilon-1}\left(\sum_{x \in S_{\pi(T)}} m(x)\right)$. This protocol satisfies $\varepsilon$-LDP by Lemma 10. Now, we claim that $\frac{e^\varepsilon+1}{e^\varepsilon-1}m(x)$ is an unbiased estimate of $\langle p, T\rangle$. Indeed, $\mathbb{E}_{\mathsf{RR}_\varepsilon, x}\left[\frac{e^\varepsilon+1}{e^\varepsilon-1}m(x)\right] = \frac{e^\varepsilon+1}{e^\varepsilon-1}\left(\frac{e^\varepsilon}{e^\varepsilon+1}\mathbb{E}_x[T(x)] - \frac{1}{e^\varepsilon+1}\mathbb{E}_x[T(x)]\right) = \mathbb{E}_x[T(x)] = \langle p, T\rangle$. Moreover, $\frac{e^\varepsilon+1}{e^\varepsilon-1}m(x)$ for $x \in S_{\pi(T)}$ are $\ell$ i.i.d. random variables with values in $\left[-\frac{e^\varepsilon+1}{e^\varepsilon-1}, \frac{e^\varepsilon+1}{e^\varepsilon-1}\right]$. We can therefore apply Hoeffding's inequality to conclude that

$$\mathbb{P}\left[|\hat{p}_T - \langle p, T\rangle| \geq \alpha\right] = \mathbb{P}\left[\left|\frac{1}{\ell} \cdot \frac{e^\varepsilon+1}{e^\varepsilon-1}\left(\sum_{x \in S_{\pi(T)}} m(x)\right) - \langle p, T\rangle\right| \geq \alpha\right]$$

$$\leq \exp\left(-\frac{\ell\alpha^2}{2\left(\frac{e^\varepsilon+1}{e^\varepsilon-1}\right)^2}\right) \leq \beta/|\mathcal{T}|,$$

where the last line follows from the fact that for $\varepsilon \in (0, 1)$, we have $(\frac{e^\varepsilon+1}{e^\varepsilon-1})^2 = \Theta(1/\varepsilon^2)$. The union bound yields $|\hat{p}_T - \langle p, T\rangle| \leq \alpha$ for all $T \in \mathcal{T}$ with probability at least $1 - \beta$, as desired. $\qquad\square$

Combining Theorem 9 and Proposition 11, we have the following.

**Theorem 12.** *Let $Q$ be a set of $k$ distributions over $\mathcal{X}$ and let $\mathcal{T}$ be a set of functionals $T \in \{-1, 1\}^{\mathcal{X}}$ with the property that, for each $q, q' \in Q$, there is some $T \in \mathcal{T}$ satisfying*

$$|\langle q - q', T\rangle| \geq \phi\|q - q'\|_1. \tag{$\star$}$$

*Then, RMDE is an $\varepsilon$-LDP algorithm that requires $m = O\left(\frac{|\mathcal{T}|\log(|\mathcal{T}|/\beta)}{\phi^2\alpha^2\varepsilon^2}\right)$ samples with the following property. For any distribution $p$, it outputs a distribution $\hat{q}$ such that with probability at least $1 - \beta$*

$$\|\hat{q} - p\|_1 \leq (1 + 2\phi^{-1})\|q^* - p\|_1 + \alpha,$$

*where $q^* := \arg\min_{q \in Q}\|q - p\|_1$ denotes the closest distribution to $p$.*

Our remaining task to prove Theorem 1 is to find a small set $\mathcal{T}$ of queries that satisfies the property ($\star$) with $\phi \geq \Omega(1)$.

## 5 The Scheffé Graph

In order to outfit RMDE with an appropriate test set $\mathcal{T}$, we will begin with the Scheffé sets and pare them down by exploiting their shared information structure.

**Definition 13.** *Given distributions $q_1, \ldots, q_k \in \Delta(\mathcal{X})$, the induced $\phi$-Scheffé graph is the digraph with vertices[5] $\binom{[k]}{2} = \{\{j, j'\} : 1 \leq j < j' \leq k\}$ and an edge $\{i, i'\} \to \{j, j'\}$ whenever*

$$|\langle \delta_{jj'}, S_{ii'}\rangle| \geq \phi\|\delta_{jj'}\|_1$$

*where $\delta_{jj'} := q_j - q_{j'}$ and $S_{ii'}$ is the signed Scheffé set from $q_i$ to $q_{i'}$.*

Now recall that a dominating set in a digraph is a subset $D$ of its vertices $V$ such that every vertex $v \in V$ either belongs to $D$ or $v \in N_{\text{out}}(D)$, namely $v$ is an out-neighbour of some vertex in $D$.

Since $|\langle \delta_{jj'}, S_{jj'}\rangle| = \|\delta_{jj'}\|_1$ by Lemma 7, then clearly for any dominating set $D$ in the $\phi$-Scheffé graph, $\mathcal{T} := \{S_{jj'} : \{j, j'\} \in D\}$ will satisfy condition ($\star$) of Theorem 9, so our main goal in this section is to demonstrate the existence of a small dominating set. In particular, we show that the $1/6$-Scheffé graph for any set of $k$ distributions has domination number $\tilde{O}(k^{3/2})$.

---

[5]More generally, we write $\binom{X}{t} := \{A \subseteq X : |A| = t\}$ for shorthand.

**Theorem 14.** *For $\phi = 1/6$ and any distributions $q_1, \ldots, q_k$, the induced $\phi$-Scheffé graph has domination number at most $4k^{3/2}\sqrt{\log k}$. Moreover, there exists a randomized preprocessing algorithm that finds a dominating set of this size in $O(k^{5/2}\sqrt{\log k})$ expected time.*

Note that this bound may be loose. We ran simulations for randomly selected $Q$ and observed weak empirical evidence that the domination number behaves as $\tilde{O}(k)$. This is because, for small values of $k$ (namely $< 20$), the Scheffé graph appears to be much denser than the following analysis suggests.

In any case, the first step to proving the bound is to examine the structure of a fixed *triangle* $\{\{j, j'\}, \{j', j''\}, \{j, j''\}\}$. We will argue that, if there are no vertices whose corresponding $\delta$ has small $\ell_1$-length (relative to the other vertices), then each vertex sends an edge to at least one other vertex in the triangle. On the other hand, if a vertex has very small $\ell_1$-length, then we can show that the remaining two vertices must share a bidirectional edge.

**Proposition 15** (Triangular Substructure). *For $\phi = 1/6$, the induced $\phi$-Scheffé graph on any distributions $q_1, \ldots, q_k$ has the following triangular structure. For every $\{j, j', j''\} \in \binom{[k]}{3}$, the graph has at least one of the following edge structures:*

(i) $\{j, j''\} \leftrightarrow \{j', j''\}$

(ii) $\{j, j'\} \rightarrow \{j, j''\}$

(iii) $\{j, j'\} \rightarrow \{j', j''\}$

The argument relies on the following geometric property of a triangle in a metric space.

**Lemma 16.** *For any three points $x, y, z$ in a metric space with metric $d$, let $a := d(x, y)$, $b := d(x, z)$, and $c := d(y, z)$ denote the lengths of each leg of the triangle $xyz$. Then either*

$$a \leq \frac{1}{2}b \quad and \quad a \leq \frac{1}{2}c \qquad or \qquad a > \frac{1}{3}b \quad and \quad a > \frac{1}{3}c.$$

To prove the lemma, assume the first condition fails, namely $a > \frac{1}{2}b$ or $a > \frac{1}{2}c$. If $a > \frac{1}{2}b$, then certainly $a > \frac{1}{3}b$ and, by the triangle inequality,

$$a > \frac{1}{2}b \geq \frac{1}{2}(c - a) \implies \frac{3}{2}a > \frac{1}{2}c \implies a > \frac{1}{3}c.$$

The case $a > \frac{1}{2}c$ is analogous.

*Proof of Proposition 15.* For simplicity, let $j = 1$, $j' = 2$, and $j'' = 3$. With an eye toward the geometric lemma, assume first that $\delta_{12}$ is "short", i.e.

$$\|\delta_{12}\|_1 \leq \frac{1}{2}\|\delta_{13}\|_1, \frac{1}{2}\|\delta_{23}\|_1.$$

In this case, since $\delta_{23} = \delta_{13} - \delta_{12}$, the triangle inequality yields

$$\|\delta_{23}\|_1 = |\langle \delta_{23}, S_{23}\rangle| \leq |\langle \delta_{13}, S_{23}\rangle| + |\langle \delta_{12}, S_{23}\rangle| \leq |\langle \delta_{13}, S_{23}\rangle| + \|\delta_{12}\|,$$

so

$$|\langle \delta_{13}, S_{23}\rangle| \geq \|\delta_{23}\|_1 - \|\delta_{12}\|_1 \geq \frac{1}{2}\|\delta_{23}\|_1 \geq \frac{1}{2}(\|\delta_{13}\|_1 - \|\delta_{12}\|_1) \geq \frac{1}{4}\|\delta_{13}\|_1$$

and thus we have an edge $\{2, 3\} \rightarrow \{1, 3\}$. By symmetry we also have an edge $\{1, 3\} \rightarrow \{2, 3\}$.

On the other hand, by Lemma 16, the remaining case to consider is that

$$\|\delta_{12}\|_1 > \frac{1}{3}\|\delta_{13}\|_1, \frac{1}{3}\|\delta_{23}\|_1.$$

Then, since $\delta_{12} = \delta_{13} - \delta_{23}$, we have

$$\|\delta_{12}\|_1 = |\langle \delta_{12}, S_{12}\rangle| \leq |\langle \delta_{13}, S_{12}\rangle| + |\langle \delta_{23}, S_{12}\rangle|.$$

By averaging, either $|\langle \delta_{13}, T_{12}\rangle| \geq \frac{1}{2}\|\delta_{12}\|_1 > \frac{1}{6}\|\delta_{13}\|_1$, so we have an edge $\{1, 2\} \rightarrow \{1, 3\}$, or $|\langle \delta_{23}, S_{12}\rangle| \geq \frac{1}{2}\|\delta_{12}\|_1 > \frac{1}{6}\|\delta_{23}\|_1$, in which case we get an edge $\{1, 2\} \rightarrow \{2, 3\}$. □

The consequence of this triangular substructure is that the graph must have relatively dense edges and thus only relatively few vertices can be supported by a small number of other signed Scheffé sets.

**Proposition 17.** *For any $r \geq 1$, the $1/6$-Scheffé graph on any $k$ distributions has at most $3kr$ vertices with in-degree less than $r$.*

*Proof.* Suppose not. By averaging over the following covering of the vertex set

$$V = V_1 \cup \cdots \cup V_k$$

where $V_j := \{v \in V : j \in v\}$, there must be some $j$ for which $V_j$ contains at least $3r$ vertices with in-degree less than $r$. Call these vertices $B_j$. Now, for any pair of vertices $\{j, j'\} \neq \{j, j''\}$ in $B_j$, either $\{j, j'\} \leftrightarrow \{j, j''\}$, $\{j', j''\} \rightarrow \{j, j'\}$, or $\{j', j''\} \rightarrow \{j, j''\}$ by Proposition 15. That is, for each pair of vertices $v \neq v'$ in $B_j$, there is at least one corresponding edge that lands in $B_j$, so

$$\sum_{v \in B_j} d_{\mathrm{in}}(v) \geq \binom{|B_j|}{2} = \frac{1}{2}|B_j|(|B_j| - 1).$$

By averaging again, there must be some $v \in B_j$ for which

$$d_{\mathrm{in}}(v) \geq \frac{1}{2}(|B_j| - 1) \geq \frac{1}{2}(3r - 1) \geq r,$$

which is a contradiction. $\qquad\square$

*Proof of Theorem 14.* We proceed by dominating vertices with small in-degree separately from vertices with large in-degree.

To that end, set $r := \sqrt{k \log k}$ and let $B$ be those vertices with in-degree less than $r$. By the previous proposition, $|B| \leq 3k^{3/2}\sqrt{\log k}$.

Now, draw uniformly at random a subset $R$ of size $\ell := k^{3/2}\sqrt{\log k}$ from the whole vertex set $V$. For a fixed $v \in V \setminus B$,

$$\mathbb{P}(v \notin N_{\mathrm{out}}(R)) = \frac{\binom{|V| - d_{\mathrm{in}}(v)}{\ell}}{\binom{|V|}{\ell}} = \left(\frac{|V| - d_{\mathrm{in}}(v)}{|V|}\right)\left(\frac{|V| - d_{\mathrm{in}}(v) - 1}{|V| - 1}\right)\cdots\left(\frac{|V| - d_{\mathrm{in}}(v) - \ell + 1}{|V| - \ell + 1}\right)$$

$$\leq \left(\frac{|V| - d_{\mathrm{in}}(v)}{|V|}\right)^{\ell} \leq 2^{-d_{\mathrm{in}}(v)\ell/|V|} \leq 2^{-2\log k} = \frac{1}{k^2}.$$

By the union bound, $\mathbb{P}(V \setminus B \not\subseteq N_{\mathrm{out}}(R)) \leq \sum_{v \in V \setminus B} \mathbb{P}(v \notin N_{\mathrm{out}}(R))) \leq |V|/k^2 \leq 1/2$, so, by the probabilistic method, there must be some $R \in \binom{V}{\ell}$ that dominates $V \setminus B$, in which case $D = B \cup R$ is a dominating set of size at most $|B| + \ell \leq 4k^{3/2}\sqrt{\log k}$.

As for finding such a dominating set algorithmically, pick $R \in \binom{V}{k^{3/2}\sqrt{\log k}}$ uniformly at random. Iterate over $v = \{a, b\} \in R$, add $v$ and its triangular out-neighbours to a hashtable in $O(k)$ time[6] by checking $\{a, i\}$ and $\{b, i\}$ for each $i \in [k] \setminus v$. By the preceding calculation, $R$ together with all uncovered vertices in the hashtable is a dominating set of size at most $4k^{3/2}\sqrt{\log k}$ with probability greater than $1/2$, so repeating until this is the case achieves the desired expected runtime. $\qquad\square$

## 6 Barriers to a Near-Linear Algorithm

The ideal algorithm for non-interactive locally private hypothesis selection would require only $\tilde{O}(k)$ samples, matching known lower bounds for the problem [DR19, GKK$^+$20]. In this section, we rule out two different approaches one could employ to design such an algorithm.

---

[6]We treat the time to check whether $|\langle q_j - q_{j'}, S_{ii'}\rangle| \geq \phi \|q_j - q_{j'}\|_1$ as a single unit of computation. In practice, this requires computing a sum or integral and depends on how the distributions are stored in memory.

## 6.1 An $\tilde{\Omega}(k^{3/2})$ Lower Bound under Triangular Substructure Assumption

One could conceive of a strengthening of Theorem 14, which argues that the domination number of any Scheffé graph is $\tilde{O}(k)$. Unfortunately, we argue that the triangular substructure described in Proposition 15 is insufficient to yield a better bound than $\tilde{O}(k^{3/2})$. Therefore, to go beyond this bound, one must employ additional structure of the Scheffé graph.

**Theorem 18.** *For all sufficiently large $k$, there is a digraph $G_k$ on vertices $\binom{[k]}{2}$ satisfying the triangular substructure condition of Proposition 15 for which*

$$dom(G_k) \geq \frac{k^{3/2}}{8\sqrt{\log k}} = \tilde{\Omega}(k^{3/2}).$$

*Proof.* Draw uniformly at random a set $R$ of size $\ell := \frac{1}{4}k^{3/2}\sqrt{\log k}$ from the vertex set $V = \binom{[k]}{2}$. For a fixed vertex $v = \{a, b\} \in V$, consider the set of indices

$$I_v^R := \{i \in [k] \setminus v : \{a, i\}, \{b, i\} \in R\}$$

that form a triangle with $v$ in which both vertices other than $v$ land in $R$. We aim to show that this set is small with high probability. Indeed, setting $t := 2\log k$, the union bound yields

$$\mathbb{P}(|I_v^R| \geq t) \leq \sum_{J \in \binom{[k] \setminus v}{t}} \mathbb{P}(\{\{a, i\} : i \in J\} \cup \{\{b, i\} : i \in J\} \subseteq R) = \binom{k-2}{t} \cdot \frac{\binom{|V|-2t}{\ell-2t}}{\binom{|V|}{\ell}}$$

$$\leq \left(\frac{e(k-2)}{t}\right)^t \left(\frac{\ell}{|V|}\right)^{2t} = \left(\frac{e(k-2)}{2\log k} \cdot \frac{\frac{1}{16}k^3\log k}{\frac{1}{4}k^2(k-1)^2}\right)^t = \left(\underbrace{\frac{e}{8}}_{\leq 1/2} \cdot \underbrace{\frac{k(k-2)}{(k-1)^2}}_{\leq 1}\right)^t$$

$$\leq 2^{-2\log k} = 1/k^2.$$

By another union bound, we can show that all such sets are likely to be small simultaneously, i.e. $\mathbb{P}(\exists v \in V, |I_v^R| \geq t) \leq |V|/k^2 \leq 1/2$, so there must be some $R \subseteq V$ of size $\ell = \frac{1}{4}k^{3/2}\sqrt{\log k}$ for which all $I_v^R$ have size less than $t = 2\log k$.

We can now form the bad digraph $G_k$ on the vertex set $V$ by adding edges as follows. For every $v = \{a, b\} \in V$ and every $i \in [k] \setminus v$,

$$\{a, i\} \in R, \{b, i\} \notin R \implies \{a, b\} \to \{b, i\}$$
$$\{a, i\} \notin R, \{b, i\} \in R \implies \{a, b\} \to \{a, i\}$$
$$\text{otherwise} \implies \{a, b\} \to \{a, i\} \text{ or } \{a, b\} \to \{b, i\} \text{ arbitrarily.}$$

Clearly, every triangle of $G_k$ satisfies either condition (ii) or (iii) of Proposition 15. Moreover, for a vertex $v = \{a, b\}$, it can only dominate an element $\{a, i\}$ or $\{b, i\}$ of $R$ if both $\{a, i\}$ and $\{b, i\}$ belong to $R$, namely $i \in I_v^R$. Therefore, including itself, $v$ dominates at most $|I_v^R| + 1 \leq 2\log k$ elements of $R$, so any dominating set must have size at least $\frac{|R|}{2\log k} = \frac{k^{3/2}}{8\sqrt{\log k}}$. $\qquad\square$

## 6.2 A Counterexample to Flattening

Another possible technique for non-interactive LDP hypothesis selection is that of *flattening*, which is discussed in [GKK+20]. The following conjecture (Question 4.4 in that work), states that any collection of distributions over a finite domain can be mapped to distributions close to uniform while still preserving their pairwise $\ell_1$-distances. Note that the original conjecture contained minor mistakes as it was written—including a missing factor of two—which we have corrected.

**Conjecture 19** (Flattening)**.** *Let $q_1, \ldots, q_k \in \Delta([n])$ be distributions that are separated in $\ell_1$-distance by at most $2\alpha$. Then there exists a randomized map $\phi : [n] \to [m]$ satisfying*

    *1. For all $1 \leq j \leq k$ and $y \in [m]$, $\frac{1-\alpha}{m} \leq \phi q_j(y) \leq \frac{1+\alpha}{m}$ and*

2. For all $1 \leq j < j' \leq k$, $\|\phi q_j - \phi q_{j'}\|_1 \in \Theta(\|q_j - q_{j'}\|_1)$

*where $\phi q$ means the distribution of $\phi(x)$, $x \sim q$.*

If the conjecture is true, then one can in effect compare any two distributions $q_j$ and $q_{j'}$ by applying the mapping $\phi$ to each and then comparing them separately to the uniform distribution $\mathcal{U}([m])$. In this case, only $k$ comparisons to the intermediary uniform distribution are required to gain information about all $\binom{k}{2}$ pairwise comparisons. These comparisons can be carried out in parallel with a small number of samples each, leading to non-interactive LDP hypothesis selection. For more details see the proof of Lemma 4.1 in [GKK+20]. Unfortunately, we show that this conjecture is false.

*Counterexample to Flattening Conjecture.* For simplicity, we identify a distribution with its mass function as a column vector in $\mathbb{R}^n$ or $\mathbb{R}^m$ and a sequence of distributions $(q_1, \ldots, q_k)$ with the matrix $Q$ whose columns are $q_1, \ldots, q_k$. Similarly, we identify a stochastic map $\phi$ with a left stochastic matrix (LSM) in $\mathbb{R}^{m \times n}$ so that $(\phi q_1, \ldots, \phi q_k)$ is just a matrix multiplication $\phi(q_1, \ldots, q_k)$

Consider the $n \times n$ identity matrix $E := I_n$. We construct $n$ additional distributions that, for the purposes of flattening, will conflict with the columns of $E$. To that end, let $H$ denote the $n \times n$ Hadamard matrix, i.e., $H_{ij} := (-1)^{\langle i,j \rangle \bmod 2}$ where $i$ and $j$ are viewed as binary strings of length $\log n$. Key properties of this matrix are that $H/\sqrt{n}$ is orthonormal, every pair of columns differs in exactly $n/2$ entries, and every column sums to $0$, except for the first column which is all ones. Now, let $F$ be an $n \times n$ matrix whose first column $f_1$ is the uniform distribution $\mathbf{1}/n$ and whose $j^{\text{th}}$ column $f_j$ is the $j^{\text{th}}$ column of $H$ with $-1$ replaced by $0$ and $+1$ replaced by $2/n$. In this case, $Q = (E, F) \in \mathbb{R}^{n \times k}$ ($k := 2n$) consists of distributions separated in $\ell_1$-distance by at most $2$.

Now, assume we have an LSM $\phi \in \mathbb{R}^{m \times n}$ satisfying the first condition $\phi Q \in [0, 2/m]^{m \times k}$. In particular, all entries of $\phi = \phi E$ must fall in $[0, 2/m]$. On the other hand, by construction we have $H = n(f_1, f_2 - f_1, \ldots, f_n - f_1)$, so, since $\frac{1}{\sqrt{n}}H$ is orthonormal, we have

$$\|\phi(f_1, f_2 - f_1, \ldots, f_n - f_1)\|_F^2 = \|\phi H/\sqrt{n}\|_F^2/n = \|\phi\|_F^2/n \leq mn(2/m)^2/n = 4/m.$$

By averaging, there must be some $v \in \{f_1, f_2 - f_1, \ldots, f_n - f_1\}$ for which

$$\|\phi v\|_1^2 \leq m\|\phi v\|_2^2 \leq m\|\phi(f_1, f_2 - f_1, \ldots, f_n - f_1)\|_F^2/n \leq 4/n.$$

Provided that $n > 4$, this is impossible for $v = f_1$ because $\|\phi f_1\|_1 = 1$, so there must be $1 < i \leq n$ such that $\|\phi f_i - \phi f_1\|_1 \leq 2/\sqrt{n} = o(1) = o(\|f_i - f_1\|_1)$. $\qquad\square$

## 7  Conclusion

In this work we introduce two new techniques for hypothesis selection.

The first is a relaxation of the classical minimum distance estimator in which the Scheffé sets are replaced by any collection of queries that is diverse enough for $\ell_1$-comparisons between any pair of candidate distributions.

The second is a new object called the Scheffé graph that contains structural information about the relationship between queries a hypothesis selection algorithm might ask. Our analysis of the Scheffé graph reveals a dense triangular substructure that can be exploited to yield a non-trivial reduction in query complexity. We show that our analysis of query complexity arising from the triangular substructure is nearly tight, so any further reduction in query complexity via the Scheffé graph will require the discovery of additional graph substructure.

Combining these two techniques yields an algorithm for non-interactive hypothesis selection under LDP constraints with state-of-the-art sample complexity, though we stress that our techniques are relevant to hypothesis selection problems more broadly.

## Acknowledgments

GK is supported by a Canada CIFAR AI Chair, an NSERC Discovery Grant, and an Ontario Early Researcher Award. MR is supported by an NSERC CGS-D scholarship. DW is supported by a Simons Investigator Award and Office of Naval Research award number N000142112647.

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
