# OpenReview forum: "Query-Efficient Locally Private Hypothesis Selection via the Scheffe Graph"
_NeurIPS.cc/2025/Conference — NeurIPS 2025 poster_

### Official Review · Reviewer_6GnU · 2025-06-27

**Clarity:** 4
**Significance:** 3
**Originality:** 4
**Rating:** 5
**Confidence:** 3

**Summary:**

This paper presents a new local differential privacy (LDP) mechanism for the hypothesis selection problem.

We are given $ k $ probability distributions $ q\_1, \\ldots, q\_k $.
There is an unknown distribution $ p $ from which $ n $ individuals each receive an independent sample.
The task is to design a mechanism which guarantees $ \\varepsilon $-LDP for each individual, and allows approximately determining which $ q\_i $ is closest to $ p $ in $ \\ell\_1 $ distance.

Here, "approximately" means the algorithm selects a distribution $ q\_i $ such that
$$ \\Vert q_i - p \\Vert_1 \\le 13 \min\_j \\Vert q\_j  - p \Vert_1 + \\alpha $$
where $ \\alpha > 0 $ is a tunable parameter which affects the number of samples required.

The LDP mechanism is non-interactive, and solves the *agnostic* version of the problem, meaning $ p $ may be different from every $ q\_i $.

For a fixed $ \\varepsilon $ and $ \\alpha $, the algorithm requires $ \\tilde{O}( k^{ 3/2 } ) $ samples, compared to a previous best of $ O( n^2 ) $ for this variant of the problem.

(Filling in some detail: if $ \\varepsilon $ and $ \\alpha $ are allowed to vary, and the algorithm is allowed to fail with probability $ \\beta $, the number of samples required is $ \\tilde{O}( k^{ 3/2 } \\log( 1/ \\beta ) / ( \\alpha^2 \\varepsilon^2 ) $.)

The paper also describes two barriers to improving this $ \tilde{O}( k^{3/2} ) $: first, an obstacle to the particular technique they use to reduce the number of comparisons; and second, a refutation of a conjecture that would have led to another approach.

**Questions:**

I've grouped my questions and comments by importance.
Please look at the "High importance" section and at least skim "Medium importance"; the "Low importance" section is just little things I noticed as I was reading.

# High importance

The statements of Theorems 1 and 13 claim $ \\tilde{O}( k^{ 5/2 } ) $ preprocessing time, but then in footnote 3 on page 7 it's admitted this assumes certain comparisons between distributions can be done in constant time.
This feels to me like a lie even if it's close to the truth.
It would be good to be more upfront in telling the reader this extra factor is missing.

# Medium importance

- The local differential privacy scenario isn't described explicitl until line 160 ("each user holds an independent datapoint $ x \\sim p $") as far as I can tell. Not a serious problem, since it's the obvious scenario to study (instead of, I don't know, maybe each individual holds one of the distributions $ q\_i $?). Still, I think it would be good to say it explicitly earlier.

- Lines 30-32: what notion of adjacent databases were those papers assuming when they showed $ O( \\log k ) $ samples suffice? (Or, almost equivalently: what does one user contribute to the database?) Worth mentioning in my opinion, unless you are really having trouble with the page limit.

- Theorem 1: $ \\alpha $ is missing a logical quantifier. For example one way to resolve this would be to add "for every $ \alpha > 0 $" at the end of line 61, and then perhaps on line 62 replace $ n \\ge n\_0 $ with $ n \\ge n\_0( \\alpha ) $. Maybe there's a better way to write it. (Now that I write this, maybe it's better the way it's written now, even if it's a bit imprecise.)

- Line 157: "...with small probability": the probability is only small if the LDP parameter $ \\varepsilon $ is large. I personally wouldn't write "small" here since I think the best LDP setting would be something like $ \\varepsilon = 1 $ so user privacy is well-protected.

- Line 167: "Let $ \\ell = |S| / | \\mathcal T | $ ...": $S$ is not defined at this point. (The next sentence makes it clear $S$ is the sample.)

- Some notes about the proof of Theorem 17:

	- It talks about a set $S$. I think you mean $R$, unless I've missed something. (At least: the equation between lines 264-265, then line 265 itself, then line 274 in $ I\_v^S $.

	- Consider ensuring $ t $ is an integer; $ 2 \\log k $ might not be. But maybe this is not worth the notational clutter. (Similarly, you could clarify the base of the logarithm is 2, but again, that's more clutter.)

	- In the display math line after line 266, I think $ \\binom{ k-2 }{ t } $ should be $ \\binom{ k-1 }{ t } $, and that change would propagate through the math that follows.

	- I found the inequality $ \\binom{ k-2 }{ t } \\cdot \\frac{ \\binom{ |V|-2t }{ \\ell-2t } }{ \\binom{ |V| }{ \\ell } } \\le \\left( \\frac{ e (k-2) }{ t } \\right)^t \\left( \\frac{ \\ell }{ |V| } \\right)^{ 2t } $ tricky to verify. It might be worth elaborating. Maybe state the approximation to the factorial function you're using somewhere.

	- Line 273: "both of them" is ambiguous; could refer to $ \\{ a,i \\} $ and $ \\{ b,i \\} $, or to $ v $ and whichever of those two is being dominated. Only one interpretation makes sense, but I got stuck on this sentence for a bit and the ambiguity didn't help. (The reason I got stuck is that my eyes skipped over the words "of $R$" the first few times I read the sentence, but that's my fault.)

	- Line 274: "$v$ dominates at most $|I\_v^S|+1$ (I'm assuming $S$ means $R$): I think a factor of 2 is missing, since each $ i \\in I\_v^R $ corresponds to two vertices $ \\{a,i\\}, \\{b,i\\} $.

- Line 281: What does $ \\alpha $-separated mean? I infer it means that every pair of distributions is *at least* distance $ \\alpha $ apart, but when I was first trying to understand the conjecture I entertained the idea it might be *at most*.

- Line 264: the arXiv version of [GKK+20] says "with high probability" here. Not sure how much that matters.

- I didn't really understand the discussion on lines 286-291. Suppose we have compared each of the $k$ distributions to the uniform distribution. Then how does that help us estimate $ \\Vert q\_i - q\_j \\Vert_1 $?
We know $ \\Vert q\_i - u \\Vert_1 $ and $ \\Vert q\_j - u \\Vert_1 $ but that only lets us upper-bound $ \\Vert q\_i - q\_j \\Vert_1 $. Probably if I spent some time reading [GKK+20] the situation would become more clear, but it would be nice if this were more self-contained.

- Line 307: "Since $ \\phi f\_1 = \\mathbf{1} / m $...": I don't think that's necessarily the case. However we have $ \\Vert \\phi f_1 \Vert_1 = 1 $ which is enough.


# Low importance

- Scheffé vs. Scheffe: the latter appears in the title but the former appears elsewhere. I'm not sure why é got changed to e.

**Ethical Concerns:**

["NO or VERY MINOR ethics concerns only"]

**Final Justification:**

I was tempted to rate this paper as 6 instead of 5, but I am not really familiar enough with the area to be confident that the k^2 -> k^2/3 improvement is a groundbreaking result; it just seems interesting to me as someone who doesn't do a lot of work on local DP. (Also the description of the 6 rating includes "exceptionally strong evaluation"; I'm not sure if exceptionally strong theorems count!)

The authors addressed my concerns, except that a small empirical part would have been nice, though I think their positioning this paper as a theory paper is a reasonable response to that.

I'm maintaining my positive rating of 5 (accept).

**Limitations:**

yes

**Quality:**

4

**Strengths And Weaknesses:**

# Strength: improved non-interactive locally differentially private hypothesis selection

The method described in this paper has a significantly better sample complexity than previously known in the non-interactive agnostic setting.

# Strength: simplicity

The method sounds fairly easy to implement.

# Strength: clear presentation; careful analysis including barriers

I really enjoyed reading this paper.
It was quite clearly presented.
For example, I liked the way the proof of Theorem 1 is divided into Theorems 11 and 13, and Theorem 11 is in turn proved by Theorem 8 and Lemma 10, and I liked the proof sketch on lines 201-205.

The mathematical analysis was carefully done and includes barriers that would need to be overcome to improve the method.

# Weakness: practical applicability / evaluation?

The practical importance or usefulness of the result is not completely clear.

If there is a concrete real-world situation where someone will benefit from this (for example, if someone is already attempting to do hypothesis selection but is limited by the available methods) then that would be worth mentioning.
Or, similarly, if this variant of the problem has been a longstanding open problem getting a lot of attention, then this would also be worth mentioning.
(Even without this, the problem is generic enough that it's bound to be useful to someone, so this doesn't seem like a fatal flaw. It just doesn't feel like a paper with "groundbreaking impact".)

Similarly, it wouldn't hurt to have an experimental evaluation, even on artificial data.
Failing that, maybe just walk the reader through an example scenario with a particular $ \\varepsilon $ and set of distributions $ q\_i $ and what Theorem 1 implies about how accuracy and number of samples trade off.
This is not especially important and the paper feels complete to me even without it.

# Minor weaknesses: errors, some unclear points

There were some minor errors and a few things that were unclear or poorly stated.

One that stands out to me: the preprocessing is said to take $ \\tilde{O}( k^{ 5/2 } ) $ time, but (admitted only in footnote 3 on page 7) this doesn't account for the time to compare a pair of distributions.
The authors should be upfront about the actual runtime. (In the statements of Theorems 1 and 13.)
(Maybe in the typical setting this time is neglibile compared to $ k $? If so, that could be mentioned explicitly.)

On the whole this paper ranks well in terms of clarity and technical accuracy: this is not a weakness relative to most other papers.

---

> ### Author Rebuttal · Authors · 2025-07-30
>
> > Importance of the work
>
> Indeed, our work falls into the latter category that the reviewer points out: we address an established open problem from the work of Gopi et al. (COLT 2020), on achieving improved algorithms for non-interactive LDP  hypothesis selection.
> This paper has received 41 citations (a reasonable number for theory works in this space), including from works that directly try to understand the role of interactivity in LDP hypothesis selection (Pour et al., COLT 2024), giving further evidence towards the importance of the problem.
> Breaking the $O(k^2)$ sample barrier (and making it "half way" to $\tilde O(k)$) is a significant step forward, since as far as we are aware, all prior methods for hypothesis selection which bypass $O(k^2)$ comparisons (in both the private and non-private settings) require substantial interactivity.
>
> > Experimental evaluation
>
> While experimental evaluation would indeed be interesting, we choose to fit our work with existing work in this field, by studying theoretical properties of algorithms.
> This is established as the first-order goal in the area of hypothesis selection, given the precedent not only in the work of Gopi et al. (COLT 2020), but also several recent papers on this problem published in NeurIPS in particular, including Bun et al. (NeurIPS 2019), Aliakbarpour et al. (NeurIPS 2023), Aliakbarpour et al. (NeurIPS 2024), none of which consider experimental evaluation.
> Such investigation would be interesting, but is outside the scope of our present work.
>
> > The statements of Theorems 1 and 13 claim $\tilde{O}(k^{5/2})$ preprocessing time, but then in footnote 3 on page 7 it's admitted this assumes certain comparisons between distributions can be done in constant time. This feels to me like a lie even if it's close to the truth. It would be good to be more upfront in telling the reader this extra factor is missing.
>
> Thanks for this comment.
> Indeed, the running time of preprocessing could vary based on the precise access to the hypotheses.
> While we consider our assumption to be fairly mild, we will increase clarity by being explicit about the exact operations one can perform on the hypotheses and their respective cost.
>
> > The local differential privacy scenario isn't described explicitl until line 160 ("each user holds an independent datapoint $x \sim p$") as far as I can tell. Not a serious problem, since it's the obvious scenario to study (instead of, I don't know, maybe each individual holds one of the distributions ?). Still, I think it would be good to say it explicitly earlier.
>
> Thanks for pointing this out. We will update the discussion in Section 1 (before summary of existing results) to highlight that the LDP setting we consider is where each user holds an independent sample from the distribution $p$.
>
> > Lines 30-32: what notion of adjacent databases were those papers assuming when they showed $O(\log{k})$ samples suffice? (Or, almost equivalently: what does one user contribute to the database?) Worth mentioning in my opinion, unless you are really having trouble with the page limit.
>
> The results which achieve $O(\log k)$ hold under essentially any neighboring relation (e.g., add/remove or substitution of a single point), which is rather similar to our work.
> However, the key difference with our work (resulting in the exponential difference in the dependence on $k$) is the semantics on privacy.
> Specifically, the $O(\log k)$ works assume a trusted curator (i.e., the central DP model), who is allowed to see all the (sensitive) data in the clear.
> In contrast, our work makes much weaker assumptions, in that no trusted party exists (i.e., no trusted curator, the local DP model), which necessarily incurs a $\mathrm{poly}(k)$ cost.
>
> > Theorem 1: $\alpha$ is missing a logical quantifier. For example one way to resolve this would be to add "for every $\alpha>0$" at the end of line 61, and then perhaps on line 62 replace $n>n_0$ with $n \geq n_0(\alpha)$ . Maybe there's a better way to write it. (Now that I write this, maybe it's better the way it's written now, even if it's a bit imprecise.)
>
> Thanks for raising this point. We have implicitly required $n_0$ to depend on the values of $\alpha$ and $\beta$ and also that the guarantees hold for any $\alpha$ and $\beta$. We agree that it could be more precise if we include that the guarantee holds for any $\alpha$ and $\beta$.
>
> > Line 157: "...with small probability": the probability is only small if the LDP parameter  is large. I personally wouldn't write "small" here since I think the best LDP setting would be something like  so user privacy is well-protected.
>
> Great point, we will adjust this wording.
>
> > Line 167: "Let  ...":  is not defined at this point. (The next sentence makes it clear $S$ is the sample.)
>
> Thank you, we will make this clearer.
>
> > Some notes about the proof of Theorem 17
>
> Thank you for your very careful reading and the helpful suggestions to improve clarity.
> We will make all the suggested changes.
>
> > Line 281: What does $\alpha$-separated mean?
>
> In this case we mean that the distributions are at most $\alpha$-far in $\ell_1$-distance. However, there is a factor of $2$ missing from the conjecture as stated by Gopi et al. and it is missing from our counterexample as well. We will correct this and make a note of it.
>
> > Line 284: the arXiv version of [GKK+20] says "with high probability" here. Not sure how much that matters.
>
> We think this use of "high probability" by Gopi et al. is a small typo (in particular, it is not clear what the probability is over) and thus we have removed it in our statement.
>
> > I didn't really understand the discussion on lines 286-291...
>
> Our goal is not to estimate $ \Vert q_i - q_j \Vert_1 $ as that is already known at preprocessing time. Rather, our goal is to somehow check which of $q_i$ and $q_j$ is closer to the unknown distribution $p$ given (locally privatized) sample access to $p$.
> The reviewer is right that the connection between Conjecture 18 and sample-efficient LDP hypothesis selection is rather non-trivial.
> Unfortunately, this non-triviality makes it unclear whether we can make it self-contained without essentially reproducing the proof of Lemma 4.1 of Gopi et al.
> Instead, we we highlight the fact that this connection is not obvious and direct the interested reader to the proof of Lemma 4.1 of Gopi et al.
>
> > Line 307: "Since $\phi f_1 = \mathbf{1}/m...$": I don't think that's necessarily the case.
>
> The reviewer is correct, and we have made that change, thank you.
>
> > Scheffé vs. Scheffe
>
> Good eye! We thought about this and made the explicit choice to have the non-accented version in the title.
> This is because accented characters may be more difficult for scrapers and indexers to capture.
> It also is a bit more cumbersome to typeset in bibtex.
> To avoid all these issues, we went with the simpler non-accented version.
> There is precedence for such a decision, see, e.g., "Hyperparameter Tuning with Renyi Differential Privacy" in ICLR 2022.

---

> > ### Comment · Reviewer_6GnU · 2025-08-06
> >
> > Thank you for addressing all of my comments. I have no further concerns.

---

### Official Review · Reviewer_KcG9 · 2025-07-01

**Clarity:** 3
**Significance:** 3
**Originality:** 3
**Rating:** 5
**Confidence:** 4

**Summary:**

The authors propose a new non-interactive locally differentially private (LDP) algorithm for hypothesis selection. The algorithm is built upon a novel concept called the Scheffé graph and is supported by a theoretical sample complexity guarantee of $\widetilde{O}(k^{3/2})$, which significantly improves upon the existing bound of $\widetilde{O}(k^{2})$.

**Questions:**

Do the authors have any simulation results regarding the size of the dominating set of the Scheffé graph?

**Ethical Concerns:**

["NO or VERY MINOR ethics concerns only"]

**Final Justification:**

The new private hypothesis selection method based on Scheffe graph did improved the existing method in sample complexity, however did not reached the optimal rate. This question the potential of the Scheffe graph whether the method can be optimal. The authors provided simulation that implies the Scheffe graph may be more optimal than their theoretical result. Therefore, I believe their method will have high impact in private hypothesis selection problem.

**Limitations:**

yes

**Quality:**

3

**Strengths And Weaknesses:**

Strength. 1. Theoretical analysis is rigorous and sound. 2. The sample complexity bound of LDP hypothesis selection is significantly improved. 3. The paper is clearly organized and easy to follow.

Weakenss. 1. Lack of empirical validation. The absence of experimental comparisons with existing algorithms limits the paper’s contributions to theoretical aspects. 2. Unclear potential of the Scheffé graph: The lower bound provided suggests that the $\widetilde{O}(k^{3/2})$ barrier may be inherent to Scheffé graph-based methods. However, the bound is proven for a broader class of triangular substructures, making it unclear whether Scheffé graphs can ever reach the theoretically optimal $O(k)$ complexity. It would be helpful if the authors included empirical simulations analyzing the size of the dominating set in Scheffé graphs to provide insight into its practical behavior and possible limits.

---

> ### Author Rebuttal · Authors · 2025-07-30
>
> > Lack of empirical validation. The absence of experimental comparisons with existing algorithms limits the paper’s contributions to theoretical aspects.
>
> Indeed, our work focuses on theoretical aspects of this problem.
> This is established as the first-order goal in the area of hypothesis selection, given the precedent not only in the work of Gopi et al. (COLT 2020), but also several recent papers on this problem published in NeurIPS in particular, including Bun et al. (NeurIPS 2019), Aliakbarpour et al. (NeurIPS 2023), Aliakbarpour et al. (NeurIPS 2024), none of which consider experimental evaluation.
> Such investigation would be interesting, but is outside the scope of our present work.
>
> > Potential of the Scheffé graph
>
> As the reviewer correctly identifies, we use the triangle substructure to prove a domination number bound of $\tilde O(k^{3/2})$.
> We further show that the triangle substructure alone will not allow one to do better than $\tilde \Omega(k^{3/2})$.
> This raises the question of whether additional graph structure would help go beyond, towards reaching the optimal bound of $\tilde O(k)$.
>
> Though we did not include them in the paper, to guide our theoretical investigations, we did run some simulation results for small values of $k$ (i.e., $k < 20$). We found that the Scheffé graph tends to be much denser than the bare triangular substructure would suggest and that the correct bound may in fact be $\tilde O(k)$ as a result.
> Unfortunately, there are a few computational challenges that preclude evaluation on larger $k$.
> Primarily, since the Dominating Set problem is known to be NP-complete, such simulations amount to brute force computation.
> Secondarily, the number of nodes in the Scheffé graph grows quadratically, increasing the computational burden.
> Nonetheless, we view our simulation results as some evidence towards an $\tilde O(k)$ domination number, and will include discussion to this effect.

---

> > ### Comment · Reviewer_KcG9 · 2025-08-06
> >
> > Very interesting result! The simulation seems to show the potential of the Scheffe graph.
> >
> > I thank the authors for their response. I have no further concern.

---

### Official Review · Reviewer_kHsp · 2025-07-03

**Clarity:** 2
**Significance:** 3
**Originality:** 3
**Rating:** 4
**Confidence:** 3

**Summary:**

The paper constructs and analyzes an efficient (constant approximation, roughly $k^{3/2}$ number of samples, where $k$ is the number of given distributions) and locally private algorithm for the problem of hypothesis selection. It also discusses theoretical limitations of the new technique.

**Questions:**

1.	Clarify Definition 3 in the context of the considered problem, and explain, preferably in the context of Lemmas 9 and 10, why the algorithm satisfies LPD.
2.	Address unclear points stated in detail comments above.

**Ethical Concerns:**

["NO or VERY MINOR ethics concerns only"]

**Final Justification:**

This is a paper addressing important problem, with interesting technical idea and improvement over previous papers. Still, the presentation is not ideal, and in the discussion, it was raised that the COLT 2020 paper has more results and techniques, which makes the actual improvement in this paper a bit less substantial than I thought in the beginning. Nevertheless, I keep my weak accept recommendation.

**Limitations:**

yes

**Quality:**

3

**Strengths And Weaknesses:**

Pros:

-	The problem is among fundament ML problems

-	There is a substantial (polynomial) improvement over previous work, and novel techniques (particularly, in Section 5)

Cons:

-	Similarly as in the previous work (Gopi et al COLT 2020), the concept of Local Differential Privacy is unclear, and the analysis does not refer to the definition.

-	Detail comparison with the previous work would be useful, especially since there are nearly linear algorithms with logarithmic approximation

-	Some results (e.g., Lemma 9) are not clearly referenced (are they new or otherwise should state reference to the original work producing them)

Detail comments:

Explain footnote 1: why the stated approximation factor includes the problem when p is one of the q_i ?

Differential privacy not explained when reviewing previous work (and definition is late, on page 3, after the theorem).

Definition 3 about differential privacy is unclear. In particular, each individual i must be an algorithm satisfying Definition 2? If so, what is a multi-dimensional space for individual I (how many dimensions does it have, what are they, etc.)?

Line 118: explain acronyms PMF and PDF

Line 119: \Delta(\cal{X}) not defined

Line 163: what is “workload of queries”?

Where is Lemma 9 used?

Why is LPD guaranteed by Lemma 10?

---

> ### Author Rebuttal · Authors · 2025-07-30
>
> > Explain footnote 1: why the stated approximation factor includes the problem when $p$ is one of the $q_i$?
>
> In the main text, we only mention the special case where $p$ is one of the $q_i$'s, in which case the reviewer is correct and the approximation factor is irrelevant: since $OPT = \min_{q \in Q} \|q - p\| = 0$, then multiplying by an arbitrary approximation factor will still be $0$.
> However, Gopi et al. (COLT 2020) actually prove a more general result: they show an agnostic-style guarantee (i.e., where $OPT \neq 0$), albeit with a substantially weaker approximation guarantee than what we're aiming for.
> We tried to capture this in the footnote; to further address weakness 2 of the reviewer (requesting a more precise comparison with prior work), we are revising it to the following:
>
> The footnote: We simplify for the sake of presentation: they actually show a slightly stronger result. Let $OPT = \min_{q \in Q} \|q - p\|$. Roughly speaking, their Lemma 4.1 provides a non-interactive $\varepsilon$-LDP algorithm such that, given $n = \tilde O(k/\alpha^4\varepsilon^2)$ samples, it outputs a distribution $\hat q$ such that $\|\hat q - p\| \leq O(\sqrt{\log k}) \cdot  \sqrt{OPT} + O(\alpha)$. Note that, compared to our desired $O(1) \cdot OPT$ guarantees, their result degrades quadratically in the value of OPT, and weakens as the number of hypotheses $k$ becomes large.
>
> > Differential privacy not explained when reviewing previous work (and definition is late, on page 3, after the theorem).
>
> We feel that differential privacy's technical definition is not necessary to understand the background for our work, and many readers will already have an informal familiarity with the notion. Hence, we placed it slightly later in order to not interrupt the narrative. We will provide a forward pointer to the definition in order to assist unfamiliar readers.
>
> > Definition 3 about differential privacy is unclear. In particular, each individual i must be an algorithm satisfying Definition 2? If so, what is a multi-dimensional space for individual I (how many dimensions does it have, what are they, etc.)?
>
> > Clarify Definition 3 in the context of the considered problem, and explain, preferably in the context of Lemmas 9 and 10, why the algorithm satisfies LPD.
>
> Thanks for the comments -- it would be helpful if the reviewer can clarify what parts of the definition are unclear.
> The input space $\mathcal{X}$ for each individual is an arbitrary set. The space $\mathcal{Y}$ containing the messages is arbitrary as well. We've added the sets $\mathcal{X}$ and $\mathcal{Y}$ to the definition so it is a little easier for the reader to infer the arbitrary nature of the input/message space.
> We will replace Definition 3 with the following, which is hopefully clearer.
>
> Definition 3: Suppose there are $n$ individuals, where the $i$-th individual has datapoint $X_i$.
> A protocol is non-interactive and $\varepsilon$-local differentially private if, for every $i \in [n]$, individual $i$ computes and outputs a (randomized) message $m_i(X_i)$ (where $m_i : \mathcal{X} \rightarrow \mathcal{Y}$ is a randomized function, depending only on the datapoint $X_i$ and potentially the index $i$), and $m_i$ is $\varepsilon$-differentially private. That is, for all $i \in [n]$, any $X_i, X_i' \in \mathcal{X}$, and any event $E \subseteq \mathcal{Y}$, we have that
> \[
> 	\Pr[m_i(X_i) \in E] \leq e^\varepsilon \Pr[m_i(X_i') \in E].
> \]
>
> To clarify the relation with Lemmas 9 and 10: it is not hard to show that Randomized Response (as described by Lemma 9) is $\varepsilon$-LDP since the ratio of the two output probabilities is between $e^{-\varepsilon}$ and $e^\varepsilon$, as required by Definition 3.
> The proof is omitted since it is an established result, but we will provide a reference and include the above informal discussion.
> As for Lemma 10, privacy in this case is inherited from privacy of Lemma 9, since each individual applies Randomized Response exactly once to their data. We will add a sentence making this clear as well.
>
> > Some results (e.g., Lemma 9) are not clearly referenced (are they new or otherwise should state reference to the original work producing them)
>
> Thanks for pointing this out: Randomized Response (Lemma 9) is an established mechanism in this space, and we will provide the appropriate citations (Warner '65, Evfimievski et al. '03, Kasiviswanathan et al. '08).
>
> > Line 118: explain acronyms PMF and PDF
>
> Corrected (probability mass and density function, respectively), thank you.
>
> > Line 119: $\Delta(\cal{X})$ not defined
>
> Fixed (the probability simplex over $\cal{X}$), thank you.
>
> > Line 163: what is “workload of queries”?
>
> Thanks for bringing this to our attention. We will remove the unclear phrase from the statement of the lemma and will explain the workload in the exposition.
>
> > Where is Lemma 9 used?
>
> > Why is LPD guaranteed by Lemma 10?
>
> As mentioned in a previous response, Lemma 9 is implicitly used in the proof of Lemma 10 (in terms of both its formulation and privacy guarantees), which we will edit for greater clarity. Essentially, we will re-write $y(x)$ as $RR(x)$ and include a remark that the mechanism satisfies the same LDP guarantee by the postprocessing property of DP (which we will also reference).

---

> > ### Comment · Reviewer_kHsp · 2025-08-08
> >
> > It would be good to discuss results in Gopi et al. (COLT 2020) in more details in the paper, to avoid any misunderstanding and clearly see the level of improvement.
> >
> > *We feel that differential privacy's technical definition is not necessary to understand the background for our work*
> >
> > I believe that understanding main theorem by non-security audience requires explanations of security concepts.
> >
> > Thanks for various explanations, they are quite satisfactory, except one thing: could individuals referred to in Definition 3 use shared/correlated randomness, or not, or it does not matter? And why?

---

> > > ### Author Response · Authors · 2025-08-08
> > >
> > > Thanks for your feedback! The motivation for LDP is settings where users only trust themselves. Therefore, the randomization of each message m_i(X_i) needs to be independent of the others to prevent information about X_i from leaking to other individuals. We will add the qualifier “independent” to the definition. Note that this does not preclude the individuals from using shared randomness for public and coordination purposes.

---

### Official Review · Reviewer_UrSm · 2025-07-04

**Clarity:** 3
**Significance:** 4
**Originality:** 4
**Rating:** 5
**Confidence:** 4

**Summary:**

The paper studies hypothesis selection under LDP in the non-interactive setting. The classical baseline asks all (k choose 2) pairwise signed Scheffé tests and therefore needs O(k^2) samples. The authors introduce the Scheffé graph: This is a directed graph
whose vertices correspond to possible queries that a locally private hypothesis selection algorithm may ask and whose directed edges indicate when one query gives sufficient information to answer another query. The graph is highly connected because of triangle inequalities between probabilities. They show it has a dominating set of size O(k^(3/2) log k). so it suffices to query only those
this many tests with randomized response, and the results for all other pairs can be inferred using the graph structure.

The privacy comes from the Randomized response of the signed Scheffé test. For each vertex, we will use the debiased RR to estimate the  mean, and the overall estimation error is analyzed through Hoeffding’s inequality and union bound over all tests.

The authors argue that if one relies solely on the triangle-style relations among tests, a dominating set of size O(k^(3/2)) is necessary—so the result is essentially tight within this framework.

**Questions:**

see the weaknesses above

**Ethical Concerns:**

["NO or VERY MINOR ethics concerns only"]

**Final Justification:**

The authors clarified my concerns regarding these weaknesses.

**Limitations:**

yes.

**Quality:**

3

**Strengths And Weaknesses:**

Strengths

Prior work either requires O(k^2) sample complexity or required interactivity; the jump to O(k^3/2) is significant. The paper also introduced an elegant graph-theoretic idea: it turns Scheffé tests into a domination problem, any this may inspire further reductions.

The paper also has clear presentation and proofs are sound.

Weaknesses

1. The current analysis is tied to binary‐valued, signed Scheffé functions. In practice, many model-selection tasks rely on real-valued or multi-ary statistics (e.g., log-likelihood ratios). Can the authors explain how to extend the tests beyond signed tests?  Could one swap in the Laplace or Gaussian mechanism to privatize a bounded real-valued score and still retain the dominating-set advantage—at the cost of only a constant-factor blow-up in the sample complexity?

2. The proofs track the variance introduced by randomized response, but there is also the usual sampling noise because each user contributes just one i.i.d. draw from p. Where exactly do you bound the deviation that includes both sources?

---

> ### Author Rebuttal · Authors · 2025-07-30
>
> > The current analysis is tied to binary‐valued, signed Scheffé functions. In practice, many model-selection tasks rely on real-valued or multi-ary statistics (e.g., log-likelihood ratios). Can the authors explain how to extend the tests beyond signed tests? Could one swap in the Laplace or Gaussian mechanism to privatize a bounded real-valued score and still retain the dominating-set advantage—at the cost of only a constant-factor blow-up in the sample complexity?
>
> Unfortunately, standard log-likelihood ratio tests cannot, in general, solve our problem of hypothesis selection due to the required "agnostic" guarantees (i.e., outputting a hypothesis which is constant-factor competitive with the closest one in the set of hypotheses), even in the non-private setting. Section 6.4 of Devroye and Lugosi's book "Combinatorial Methods in Density Estimation" has a more detailed discussion of this issue.
> Note that the prior work of Gopi et al. (COLT 2020) employs a type of log-likelihood ratio test to privately perform an easier variant of hypothesis selection, achieving much weaker agnostic guarantees than we require.
> Unfortunately, in Section 6.2 we show that their core "flattening" technique cannot be improved to achieve the full desired agnostic guarantees.
> It is not clear whether any type of log-likelihood ratio test can achieve constant factor agnostic guarantees for hypothesis selection (with or without privacy) and it remains an interesting direction for further study.
>
> > The proofs track the variance introduced by randomized response, but there is also the usual sampling noise because each user contributes just one i.i.d. draw from p. Where exactly do you bound the deviation that includes both sources?
>
> Lemma 10, which is the key component in our probabilistic analysis, takes into account both sources of randomness.
> In Line 170, for any functional $T$, the curator takes an average over all $\ell$ noisy data points $y(x),x\in S_{\pi(T)}$ that are sent to the the curator. In Line 172, it is proven that for each $x\in S_{\pi(T)}$, the expected value of the random variable $\frac{e^{\varepsilon}+1}{e^{\varepsilon }-1} y(x)$ is equal to $\langle p,T\rangle$. This expectation is with respect to both sources of randomness; the randomness incurred by the randomized response, and by $x\sim p$.
> In each expectation here, the sources of randomness are indicated in the subscript. The Hoeffding's inequality in  Line 174 is bounding the probability with respect to both of these sources  that the empirical estimate $\hat{p}_T$ calculated using the random variables $y(x)$ is close to their expectation, i.e., $\langle p,T\rangle$. Finally, the union bound guarantees this bounded deviation with respect to the randomized response and i.i.d. draws from the distribution for all the functionals in $\mathcal{T}$.
> We're happy to make this clearer in the final version of the paper.

---

> > ### Comment · Reviewer_UrSm · 2025-08-08
> >
> > Thank you for the clarification! I'd also appreciate that you could add some discussion related to standard log-likelihood ratio tests to the paper. I'm happy to raise my score.

---

### Decision · Program_Chairs · 2025-09-17

**Decision:**

Accept (poster)

**Comment:**

The manuscript aims to reduce the number of queries under local differential privacy to select the best fit amongst a set of hypothesised probability distributions. The theoretical results reducing sample complexity were perceived as clearly substantial [UrSm, kHsp, KcG9, 6GnU] and while an empirical evaluation was desired [KcG9, 6Gnu], the clear presentation [UrSm, 6Gnu] with sound proofs [UrSm, KcG9] and seemingly easy-to-implement method [6Gnu] makes this work also interesting as a purely theoretical exploration where empirical results could be left to future work.